# Application of Lacunarity for Quantification of Single Molecule Localization Microscopy Images

**DOI:** 10.3390/cells11193105

**Published:** 2022-10-02

**Authors:** Bálint Barna H. Kovács, Dániel Varga, Dániel Sebők, Hajnalka Majoros, Róbert Polanek, Tibor Pankotai, Katalin Hideghéty, Ákos Kukovecz, Miklós Erdélyi

**Affiliations:** 1Department of Optics and Quantum Electronics, University of Szeged, 6720 Szeged, Hungary; 2Department of Applied and Environmental Chemistry, University of Szeged, 6720 Szeged, Hungary; 3Institute of Pathology, Albert Szent-Györgyi Medical School, University of Szeged, 6725 Szeged, Hungary; 4Centre of Excellence for Interdisciplinary Research, Development and Innovation, University of Szeged, 6723 Szeged, Hungary; 5Biomedical Applications Group, ELI-ALPS Research Institute, ELI-HU Non-Profit Ltd., 6728 Szeged, Hungary; 6Department of Oncotherapy, University of Szeged, 6720 Szeged, Hungary; 7Genome Integrity and DNA Repair Group, Hungarian Centre of Excellence for Molecular Medicine (HCEMM), University of Szeged, 6728 Szeged, Hungary

**Keywords:** lacunarity, dSTORM, quantitative analysis

## Abstract

The quantitative analysis of datasets achieved by single molecule localization microscopy is vital for studying the structure of subcellular organizations. Cluster analysis has emerged as a multi-faceted tool in the structural analysis of localization datasets. However, the results it produces greatly depend on the set parameters, and the process can be computationally intensive. Here we present a new approach for structural analysis using lacunarity. Unlike cluster analysis, lacunarity can be calculated quickly while providing definitive information about the structure of the localizations. Using simulated data, we demonstrate how lacunarity results can be interpreted. We use these interpretations to compare our lacunarity analysis with our previous cluster analysis-based results in the field of DNA repair, showing the new algorithm’s efficiency.

## 1. Introduction

The spatial resolution of conventional optical microscopes is limited by the diffraction of light. In any imaging system, the image of an arbitrarily small light source is an extended blob referred to as the point spread function (PSF) [1], the size of which depends on the wavelength, the numerical aperture and the optical aberrations. In the diffraction limited case, using a clear circular aperture, the PSF is an airy pattern [2], with a central peak approximately half the wavelength in width. Below this spatial scale, images become blurred and structures cannot be resolved. In biological samples this means that most subcellular organizations and molecular complexes remain undetected by conventional microscopy methods. To overcome this diffraction barrier, several super-resolution microscopy techniques have been developed such as structured illumination microscopy (SIM) [3], stimulated emission depletion (STED) [4] and single molecule localization microscopy (SMLM) [5]. The SMLM techniques determine the positions of single emitting fluorophores with precision of an order of magnitude higher than the diffraction limit. This allows for image resolution in the scale of tens of nanometers and provides the highest resolution super-resolution method. The most prominent SMLM techniques used in biological studies include (direct) stochastic optical reconstruction microscopy ((d)STORM) [6,7], photo activated localization microscopy (PALM) [8,9], points accumulation for imaging in nanoscale topography (PAINT) [10], ground state depletion microscopy followed by individual molecule return (GSDIM) [11], and MINFLUX [12].

The quantitative analysis of SMLM datasets has gained considerable interest in recent years [13]. However, the different data format (SMLM provides the 3D coordinates of fluorescent molecules) requires new merit functions [14,15], evaluation algorithms [16,17] and visualization methods [18]. To assess the underlying structure and density of target molecules in biological samples consistently with previous studies, several quantitative analysis methods have been proposed. These methods require the introduction of new metrics which are ideally analogous to the terms used in regular optical microscopy. SMLM images are formed by localization points showing the positions of individual fluorescent molecules that are bound to the target molecules, usually proteins, by linkers. A typical arrangement for these proteins of interest is clusters. Therefore, cluster analysis methods such as DBSCAN [19] and Voronoi tessellation [20] are widely used when the size, area and composition of clusters can be directly determined from the raw localization data. The number of target molecules is not equal to the number of accepted localizations for a variety of reasons. Labelling density depends on the number of epitopes and the dye molecules on antibodies. Individual fluorophores can blink multiple times throughout the image acquisition process before they are finally bleached. Such processes can be statistically investigated by searching for the traces of single emitters in the data [21]. The fluorescent ON-state lifetime is matched to the exposure time, but due to the stochastic nature of individual blinking events, sometimes sequential frames capture the very same dye molecules and thus create several less accurate localizations instead of one of high accuracy. This problem can be rectified by the use of trajectory fitting algorithms unifying the signal of the molecule in question [22]. Colocalization cannot be determined by measuring intensity ratios. Techniques solving this problem and providing quantitative colocalization analysis for SMLM are either based on DBSCAN, where the change in localization densities defines the relation between the two channels [23], or the Voronoi analysis, where colocalization is defined using Manders’ coefficients [24]. The importance of quantitative analyses for determining the biological significance of SMLM data can be observed through the numerous novel methods that have been developed in recent years. A technique that is yet to be implemented for use in SMLM but has proven itself in material sciences could be lacunarity analysis.

Lacunarity was first introduced by Mandelbrot to describe how certain patterns fill space and provide information about texture [25]. It was mainly used in fractal analysis to distinguish between fractals with the same dimensions but with different structures. Lacunarity can characterize patterns in a flexible yet theoretically consistent manner across different scales. Lacunarity has been found to be beneficial in various scientific fields, from micro-CT analysis [26] and astronomy [27] through food chemistry [28] and geography [29] to neuroscience [30] and oncology [31]. We believe that lacunarity analysis can be a competitive method for describing the structure of nanoscale cellular structures unraveled by single molecule localization microscopy. To demonstrate the effectiveness of lacunarity analysis, we revisited two of our previous quantitative dSTORM results [21,32] where cluster analysis played a central role in the evaluation process.

The DNA in the nucleus is constantly targeted by different damaging agents derived from different sources, causing various types of damage to the genetic code. DNA double-strand breaks (DSBs) are the most deleterious lesions, and therefore they must be repaired as quickly and efficiently as possible to prevent chromosomal loss and translocation. These repair mechanisms are carried out by several DNA repair proteins forming focuses [33,34] around the DSB. The size of these foci is in the range of a hundred nanometers, and their structure is crucial for understanding the process of DNA repair. Comparative dSTORM and confocal measurements have revealed the advantages and limitations of the two microscopy methods.

In this paper we propose a new, lacunarity based quantitative analysis method that can assess the structure and homogeneity of target molecules in SMLM based datasets, providing fast and accurate information about the structure of biological samples. We demonstrate the effectiveness of our lacunarity based algorithm by comparing its results with previously conducted DBSCAN produced results in the field of DNA repair.

## 2. Materials and Methods

### 2.1. Lacunarity Calculation

The gliding box algorithm for evaluating 2D datasets was published by Allain and Cloitre in 1991 [35]. In this method, lacunarity is obtained at a certain size “ε” by placing an ε^2^ size box at every possible different position on the image. The mass of these boxes is calculated by counting the number of object pixels inside them. Lacunarity can then be calculated from the sums of the first and second moments of the box masses. As we previously mentioned, in 2D SMLM the data consist of coordinate pairs, wherefore the calculation of lacunarity must be modified accordingly. Figure 1a shows a regular SMLM dataset consisting of numerous coordinate pairs in an L by L area where “L” is the side length of the region of interest (ROI) in nanometers. To calculate lacunarity we need to redefine the mass of a box as the number of localizations inside the box. We also need to define a step size “s” with which the boxes will glide through the image. As a consequence, the smallest possible box size will be equal to “s”. This gliding can be seen in Figure 1b. As the boxes glide through the image, one can calculate the mass of each box and write the value into a matrix “BM(i,j)” as shown in Figure 1c. To speed up the calculations, the SMLM data can be pixelized into an M by M image, where
(1)$$M=\frac{L}{s},\;$$
and the pixel values are the number of localizations in each pixel. The number of boxes for a given ε in case of a square ROI is
(2)$$N\left(\varepsilon \right)={\left(M-\varepsilon +1\right)}^{2}.$$

As previously shown by Tolle in 2008 [36], lacunarity can be more efficiently calculated from the box masses as
(3)$$\Lambda \left(\varepsilon \right)=\frac{N\left(\varepsilon \right)\cdot \sum_{i,j=1}^{N\left(\varepsilon \right)}{\left(\mathrm{BM}\left(i,j\right)\right)}^{2}}{{\left(\sum_{i,j=1}^{N\left(\varepsilon \right)}\mathrm{BM}\left(i,j\right)\right)}^{2}},$$
than by creating the probability distributions for the box masses. To further increase the speed of the calculations, we implemented the idea published by Backes 2013 [37], which notes that to calculate the box mass of a neighboring box of an already calculated one, only those edges of the box need to be visited that do not overlap with the already calculated box. These changes allow our software to calculate lacunarity at a rate of 0.2 megapixels per second on an AMD Ryzen 9 3900X system for box sizes equal to the divisors of 8000. Showcasing the speed on a typical sample consisting of 1,300,000 localizations over a 10 µm by 10 µm area using a step size of 1 nm takes 341.32 s, while on this same sample the DBSCAN algorithm used in our previous studies takes 1687.92 s to complete on the same system. The runtime can be sped up by, for example, using a step size of 5 nm, which takes only 2.53 s at the cost of losing the information about box sizes smaller than 5 nm. The runtime of lacunarity analysis is proportional to the area of the sample, while DBSCAN’s runtime is proportional to the number of localizations. In dense small samples, this can allow the lacunarity analysis to be a thousand times faster than DBSCAN. In SMLM, the typical pixel size equals the achieved resolution, which is around 20 nanometers. We recommend the use of step sizes smaller or equal to this value. The full system specs, runtime analysis and step size comparison are listed in the Appendix A. To compare the lacunarity of different datasets we developed a new normalization method for lacunarity curves. This was necessary because unlike with binary images, the lacunarity value at box size one (ε = 1 pixel) will not be mathematically the same for images with the same object pixels. Our new normalization method compares the lacunarity curve of a sample (Figure 1a) to a random dataset (Figure 1d) with the same number of localizations. The lacunarity curves, calculated for the sample (black line) and for the random data (red line), are depicted in Figure 1e. Normalization is performed by calculating the relative difference of each point on the two lacunarity curves
(4)$$\mathrm{Relative}\;\mathrm{Lacunarity}\;\mathrm{Difference}\left(\varepsilon \right)=\frac{\Lambda_{\mathrm{sample}}\left(\varepsilon \right)-\Lambda_{\mathrm{random}}\left(\varepsilon \right)}{\Lambda_{\mathrm{random}}\left(\varepsilon \right)}.$$

The result, which we will call the lacunarity difference (LD) curve of the sample, can be seen in Figure 1f; such a normalization highlights the box sizes at which heterogeneity deviates as opposed to a random dataset. The box size at which the LD curve peaks shows the size where the heterogeneity of the sample is maximal. While this peak box size does not correspond to the cluster size, it can characterize the sample and its movement can give valuable information about changing sample parameters.

### 2.2. TestSTORM Simulation

Simulated test data were generated in the dSTORM simulation software called TestSTORM [38] to mimic real life samples in which target molecules form several clusters that consist of nanofoci (dense, 10–50 nm diameter clumps of localizations). To create the simulated data, 8000 frames were generated using a modified version of the inbuilt “discs pattern” generator, which randomly places epitopes in a circular area at a set density and orientation. These circular clusters were placed in a three by three grid. The nanofoci were created using several labels per epitope with higher linker lengths. The dye and acquisition parameters were left at default values, only the “Non-spec. l. dens. (1/um^3^)” and the “Number of frames” were changed. No sample drift was introduced and a Gaussian PSF was used. Each blinking event was fitted with the rainSTORM reconstruction software using the multi-Gaussian 2D analysis algorithm. The following sample parameters were evaluated at five values: the number of clusters defined as the number of clusters in the region of interest, the size of clusters defined as the radius of a single cluster, the distance of clusters defined as the distance between the center of two neighbouring clusters, the density of nanofoci defined as the number of nanofoci over a square micron area inside of the clusters, the size of nanofoci defined as the radius of a single nanofocus, the number of localizations per nanofoci defined as the number of fluorophores belonging to a single nanofocus, and nonspecific localization density defined as the number of nonspecific localizations in a cube micron volume.

### 2.3. DNA DSB dSTORM Images

The dSTORM datasets of DNA double-strand break repair research, used for demonstrating the effectiveness of our technique, were chosen from previous studies. In these studies the DSBs were artificially induced and visualized using the phosphorylated H2AX at Ser139 (referred to as γH2AX) as a double strand break marker in the nuclei of the cells [39]. The radiation treated U251, the neocarzinostatin (NCS) treated U2OS, and 4-hydroxytamoxifen (4-OHT) treated DIvA cell lines (U2OS cell line-based systems, which express and activate AsiSI homing endonuclease upon 4-OHT addition [40]) were studied in Brunner 2021 [32] and in Varga 2019 [21], respectively.

## 3. Results

### 3.1. Lacunarity Behaviour Examined through TestSTORM Simulations

The TestSTORM simulation data can be divided into three groups based on the characteristics of the parameters. In the first group, parameters related to the clusters, i.e., cluster number, cluster size and cluster distance, were analyzed. In the second group, parameters of the nanofoci, i.e., nanofocus density, nanofocus size and localizations per nanofocus, were studied. In the third group, the density of nonspecific localizations was studied. The simulation results will be discussed based on the super-resolution images and the lacunarity difference curves; the raw lacunarity curves are available in the Appendix A. For each simulation, all other parameters were kept at their base value and only the studied parameter was changed. The base values and simulated parameter ranges are shown in Table 1.

During the simulations, the five studied cluster numbers were one, three, five, seven and nine. The super-resolution images for one, five and nine clusters can be seen in Figure 2a–c. The higher the number of clusters, the more homogeneous the image becomes, therefore the lacunarity difference (LD) curve is flattened (Figure 2j). This homogenization is caused by the clusters covering a larger portion of the image evenly, where the different parts of the image become more alike to one another. The peak of the LD curve describes the box size at which the difference from a homogeneous sample is the greatest. We can observe that the peak moves towards the smaller box sizes (127 nm → 70 nm) as the number of clusters increases (Figure 2j). This means that the image is homogenized more at larger box sizes when the number of clusters is increased.

For the cluster sizes, the five settings for the radius of the circular clusters were 140 nm, 350 nm, 560 nm, 770 nm and 980 nm. The super-resolution images for 140 nm, 560 nm and 980 nm cluster sizes can be seen in Figure 2d–f. Larger clusters cover a larger part of the image evenly, thus increasing homogeneity and flattening the LD curve. The peak of the lacunarity difference curve shifts towards the smaller box sizes as the size of the clusters increases (100 nm → 53 nm) (Figure 2k). Changes in cluster size or the number of clusters results in similar trends since they increase the area that is covered by the clusters while maintaining the localization density.

For the study of cluster distances, a constant, 3 by 3 grid was used with different cluster distances. The five settings were 500 nm, 1000 nm, 1500 nm, 2000 nm and 2500 nm. The super-resolution images for 500 nm, 1500 nm and 2500 nm cluster distances can be seen in Figure 2g–i. In the denser cases of 500 nm and 1000 nm, the 560 nm radius clusters overlap and form a continuous area. This overlap decreases the size of the covered area and increases the localization density. The result for the LD curve is a sharp increase in heterogeneity and the peak moves towards the larger box sizes (143 nm and 85 nm). In the other cases where the clusters are well separated, the amplitude and position of the LD curve peaks remain the same (70 nm, 70 nm, 69 nm) (Figure 2l).

Generally describing the first category, we can say that an increase in the cluster number or size makes the image more homogeneous and shifts the LD curve peak towards the smaller box sizes. However, the cluster distance does not affect the lacunarity difference curve significantly in the case of spatially separated clusters. Overlapping clusters introduce significant inhomogeneity and shift the LD peak towards the larger box sizes.

In the second category for nanofocus density, the five studied settings were 10 nanofoci/μm^2^, 25 nanofoci/μm^2^, 40 nanofoci/μm^2^, 55 nanofoci/μm^2^ and 70 nanofoci/μm^2^. The super-resolution images for 10/μm^2^, 40/μm^2^ and 70/μm^2^ nanofocus densities can be seen in Figure 3a–c. An increase in the density of nanofoci increases the number of localizations inside the clusters. The localization density increases from 3800 localizations/μm^2^ to 7200 localizations/μm^2^, 9700 localizations/μm^2^, 11,300 localizations/μm^2^ and 12,500 localizations/μm^2^, creating a more even distribution. This effect results in an increase in homogeneity to a certain point determined by the cluster size and shape. The peak of the lacunarity difference curve slightly shifts (69 nm → 78 nm) towards the larger box sizes, indicating that the homogenization effect is greater for the smaller box sizes (Figure 3j). This is explained by the fact that the introduction of new nanofoci only affects the image in a small localized area.

Nanofocus sizes were set with the length of linker parameter in TestSTORM. The five analyzed settings were 11 nm, 33 nm, 55 nm, 77 nm and 99 nm. The super-resolution images for 11 nm, 55 nm and 99 nm nanofocus sizes can be seen in Figure 3d–f. An increase in the nanofocus size homogenizes the image in a localized area, however nanofocus size slightly affects the effective cluster size, i.e., the increase in homogeneity does not stop abruptly. The peak of the lacunarity difference curve moves (30 nm → 88 nm) towards the larger box sizes (Figure 3k). Changes in nanofocus density or nanofocus size have similar effects on lacunarity because they both affect the homogeneity of the image inside the clusters. Consequently, they have a larger effect on smaller box sizes.

The number of localizations per nanofocus setting was studied at 10, 80, 150, 220 and 290 localizations. The super-resolution images for 10, 150 and 290 localizations per nanofocus can be seen in Figure 3g–i. At 10 localizations per nanofocus, the clusters blend into the nonspecific localizations, resulting in low heterogeneity. Higher than 80 localizations per nanofocus values result in no changes in heterogeneity, because the geometry of the image does not change. This means that both the peak value and peak box size remain the same on the lacunarity difference curve, as can be seen in Figure 3l.

In the second category, an increase in the density or size of nanofoci results in an increase in homogeneity, and this effect is more pronounced at smaller box sizes pushing the peak of the lacunarity difference curve towards the larger box sizes. Changing the number of localizations per nanofocus leads to no change in geometry, thus it has no effect on the lacunarity difference curve when the signal to noise ratio is adequate.

Finally, the density of nonspecific localizations was studied at 0/μm^3^, 70/μm^3^, 140/μm^3^, 280/μm^3^ and 560/μm^3^. The super-resolution images for 0/μm^3^, 140/μm^3^ and 560/μm^3^ nonspecific localization densities can be seen in Figure 4a–c. Higher densities of nonspecific localizations evenly homogenize the whole image, while the peak of the curve only moves very slightly (69 nm → 64 nm) towards the smaller box sizes (Figure 4d).

One can draw conclusions about the underlying mechanisms and changing parameters by evaluating the shift of the LD curve peak both in height and position. As can be seen in Figure 4e, each different parameter moves the peak of the LD curve on different curves in the relative lacunarity difference-box size space. While a certain parameter would be hard to isolate, the different parameter groups (cluster and nanofocus) are distinct enough to be separated. These changes are not specific to the cluster shapes we have chosen. In the case of structures of other shapes consisting of substructures (like clusters consisting of nanofoci) the increase of the covered area, either through a rise in structure size or structure number, would result in similar homogenization. The distance of the structures would not affect lacunarity until an overlap is apparent. The effect of substructures increasing in density or size would homogenize the structures’ interior, while the number of localizations per substructure would have a noticeable effect neither on geometry nor lacunarity. The increase in nonspecific localization density always homogenizes the image. These conclusions can be derived from the properties of lacunarity and how it is calculated.

### 3.2. Dose Dependent Lacunarity Study of the X-ray Radiation Treated Cell Line

For the biological samples we selected three datasets to study different aspects of the DSB repair mechanism. In the first one we investigated the nuclei of cells that were treated with different dose levels of X-ray radiation. For the second set we studied how the formation of DSB foci change in time after the radiation event. In the third set the effects of different DSB inducing chemical agents were analyzed. As we have previously, we will only show the super-resolution images of the area of the nuclei that the lacunarity analysis was executed on, as well as the lacunarity difference curves. The lacunarity curves are available in the Appendix A. The box size for the highest lacunarity difference value of the average curve (peak box size) and the box size of the center of mass of the lacunarity difference curve (center box size) are highlighted on each graph.

Regarding the radiation dosage we chose the 0 Gy (or control), 2 Gy and 5 Gy cell groups that we observed 30 min after they were subject to radiation induced damage. The higher the dosage the higher the expected number of DSBs, which results in more repair focus. The super-resolution images of the representative areas from each dose group can be seen in Figure 5a,b. We can observe in Figure 5d–f that the lacunarity difference curve is flattened by higher radiation doses. The images become more homogeneous as the foci get more numerous. The peak of the lacunarity difference curve moves towards the smaller box sizes with an increase in homogeneity. This indicates that the change induced by the higher radiation doses occurs at the scale of the foci and not at the scale of nanofoci. As we have shown for simulation data, this behaviour is caused by the clusters increasing in either number or size.

### 3.3. Kinetic Lacunarity Study of the X-ray Radiation Treated Cell Line

Time dependent studies were performed on the cells subjected to 5 Gys of radiation at 30 min, 24 h and 72 h after the treatment. The super-resolution images of the representative areas from each time group can be seen in Figure 6a,b. Figure 5d–f shows that the lacunarity difference curve peaks up as the DSBs are repaired and the nuclei become more like the control. From the peak moving towards the larger box size as heterogeneity increases, we can conclude that the number and size of the focuses are decreasing as the repair is done. We see that while the peak box size returns to the control value, heterogeneity ends up higher after the radiation dose and repair. This means that the nuclei may have undergone permanent changes.

### 3.4. Lacunarity Study of Chemically Treated Cell Lines

The chemically treated cells consisted of U2OS control, U2OS treated with neocarzinostatin (NCS) and DIvA treated with 4-hydroxytamoxifen (4-OHT). Super-resolution images of representative areas from each treatment group can be seen in Figure 7a–c. Slight homogenization was observed for the NCS treated U2OS cells, but the most dominant change in the treatments was the shift of the peak of heterogeneity, as can be seen in Figure 6d–f. Compared to the control, in both treatments the heterogeneity decreased at the larger box sizes while it increased at the smaller box sizes, resulting in the peak being pushed significantly towards the smaller box sizes. This phenomenon indicates that the number and size of the foci increased, while the size or density of the nanofoci decreased.

## 4. Discussion

Using the gliding-box method for lacunarity calculation, we have developed an algorithm capable of quantitative homogeneity assessment in SMLM data. Using simulated and measured datasets we have demonstrated how the lacunarity data can be interpreted and used for describing the structure of localizations in a quantitative manner. The quantitative analysis of SMLM data has always posed a challenge computationally because of the large datasets. Our lacunarity algorithm is capable of providing accurate information about the structure of SMLM data faster than previous techniques, such as cluster analysis. The use of a new visualization method, where the lacunarity curve is compared against the curve of a random dataset, enables us to instantly assess the relative homogeneity of an image at each studied box size. The previously conducted DBSCAN based cluster analyses shown in Varga 2019 [21], Figure 2 as well as Brunner 2021 [32], Figure 5 and Figure 6 are in perfect agreement with our results. The effects of radiation and DSB inducing chemical agents were shown to cause an elevation in the number of DSBs and the formation of larger repair foci. Our algorithm was capable of revealing these changes shown previously but significantly faster.

There are, however, a few important limitations to our lacunarity analysis method. The most obvious one is that any lacunarity based algorithm describes the structure of a sample at a given size by only using a single number. This results in different processes having similar effects on lacunarity. For example, in our simulation data an increase in cluster size or cluster number had a very similar effect on the lacunarity curve. This can make interpreting lacunarity results difficult without a preliminary simulation analysis. We recommend the proposed lacunarity analysis algorithm for preliminary investigations in biological studies where it can provide fast and accurate information about the structure of samples, i.e., sample screening.

## Figures and Tables

**Figure 1 cells-11-03105-f001:**
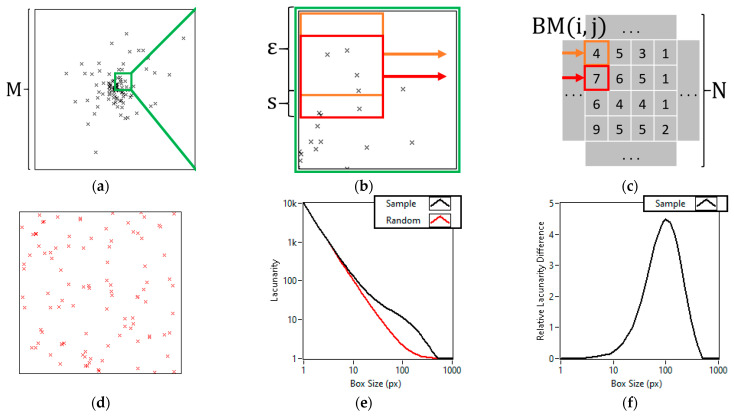
Step by step representation of the lacunarity calculation process and visualization. Localization cluster of a hundred localizations in a Gaussian distribution of 150 nm sigma on an area of 1000 nm by 1000 nm (**a**). Zoomed in on the cluster showing the gliding of an “ε” sized box at a step size of “s” (**b**). The box masses in the zoomed in area, calculated from the number of localizations in each box (**c**). Random set of a hundred localizations on the same area of 1000 nm by 1000 nm with an even distribution (**d**). Lacunarity curves of the cluster and the random dataset (**e**). Lacunarity difference curve created by calculating the relative difference between the lacunarity curve of the cluster and that of the random dataset of the same number of localizations at each point (**f**).

**Figure 2 cells-11-03105-f002:**
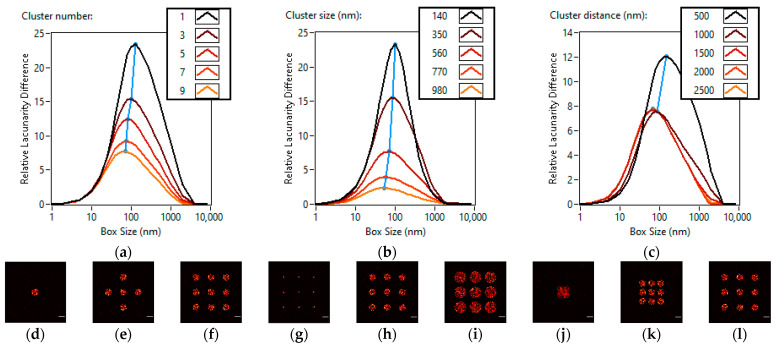
Effects of changing cluster parameters on lacunarity. Lacunarity difference curves of TestSTORM generated datasets of different cluster numbers (**a**), cluster sizes (**b**), cluster distances (**c**). Three super-resolution images of the simulated data are also shown for each lacunarity difference curve. Cluster numbers of one (**d**), five (**e**) and nine (**f**). Cluster sizes of 140 nm (**g**), 560 nm (**h**) and 980 nm (**i**). Cluster distances of 500 nm (**j**), 1500 nm (**k**) and 2500 nm (**l**). Scale bars are 1 µm.

**Figure 3 cells-11-03105-f003:**
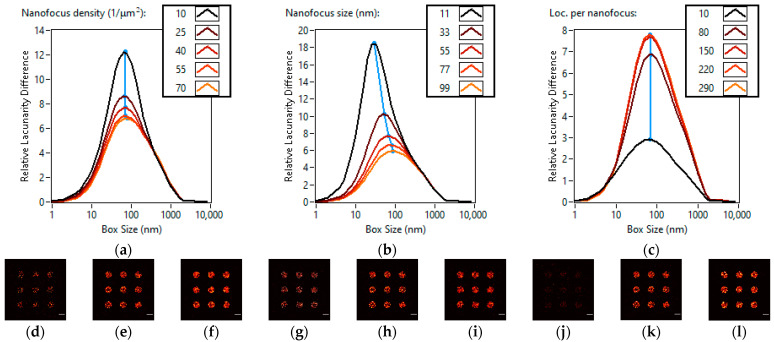
Effects of changing nanofocus parameters on lacunarity. Lacunarity difference curves of TestSTORM generated datasets of different nanofocus densities (**a**), nanfocus sizes (**b**), localizations per nanofocus (**c**). Three super-resolution images of the simulated data are also shown for each lacunarity difference curve. Nanofocus densities of 10/μm^2^ (**d**), 40/μm^2^ (**e**) and 70/μm^2^ (**f**). Nanofocus sizes of 11 nm (**g**), 55 nm (**h**) and 99 nm (**i**). Number of localizations per nanofocus of 10 (**j**), 150 (**k**) and 290 (**l**). Scale bars are 1 µm.

**Figure 4 cells-11-03105-f004:**
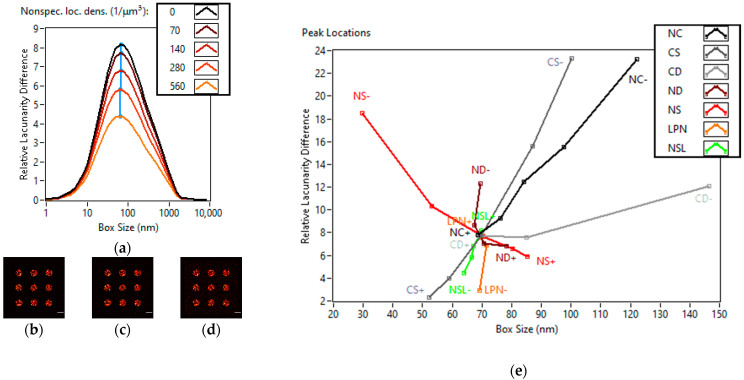
Effects of changing nonspecific localization densities on lacunarity and summary of the simulation results. Lacunarity difference curves of TestSTORM generated datasets of different nonspecific localization densities (**a**). Three super-resolution images of the simulated data with nonspecific localization densities of 0/μm^3^ (**b**), 140/μm^3^ (**c**) and 560/μm^3^ (**d**) are also shown. Scale bars are 1 µm. The peak positions for each simulation in order (**e**). Number of clusters (NC), cluster size (CS), cluster distance (CD), nanofocus density (ND), nanofocus size (NS), localizations per nanofocus (LPN) and nonspecific localization density (NSL).

**Figure 5 cells-11-03105-f005:**
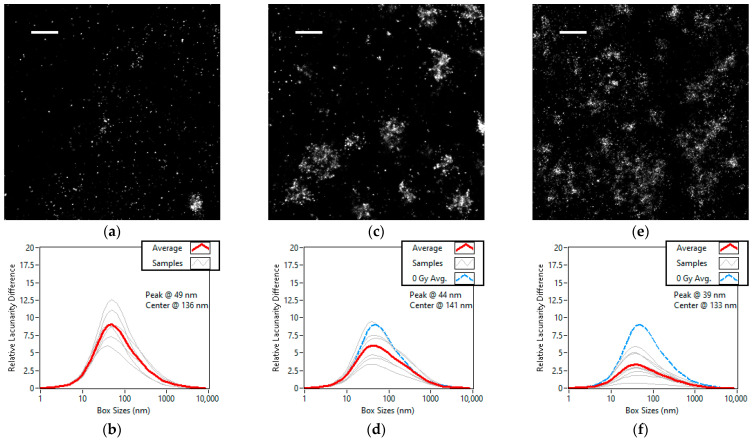
Super-resolution images and lacunarity difference curves of Alexa Fluor 647 labelled γH2AX in the nuclei of U2OS cells grouped by radiation dosage. Each cell was observed 30 min after being subjected to 0 Grays (**a**,**b**), 2 Grays (**c**,**d**) and 5 Grays (**e**,**f**) of ionizing radiation. Scale bars are 1 µm. The number of studied cells is 6, 7 and 9, respectively. On each lacunarity difference curve, the average of the curves for each cell is shown in red and the average of the untreated U2OS is shown in dashed blue.

**Figure 6 cells-11-03105-f006:**
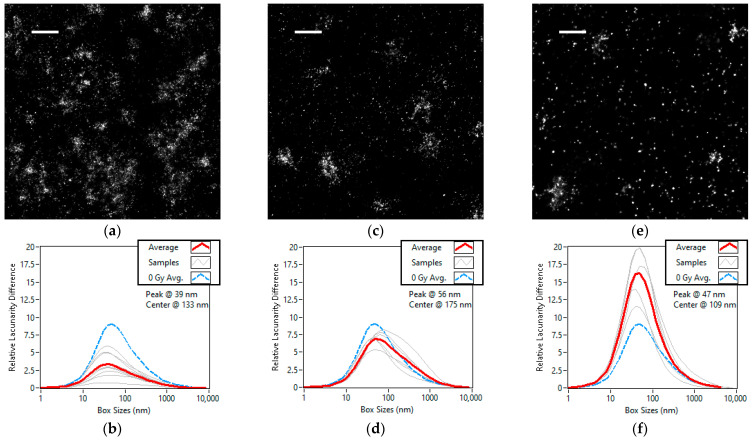
Super-resolution images and lacunarity difference curves of A647 labelled γH2AX in the nuclei of U2OS cells grouped by time after treatment. The cells were observed 30 min (**a**,**b**), 24 h (**c**,**d**) and 72 h (**e**,**f**) after being subjected to 5 Grays of ionizing radiation. Scale bars are 1 µm. The number of studied cells is 9, 6 and 5, respectively. On each lacunarity difference curve, the average of the curves for each cell is shown in red and the average of the untreated U2OS is shown in dashed blue.

**Figure 7 cells-11-03105-f007:**
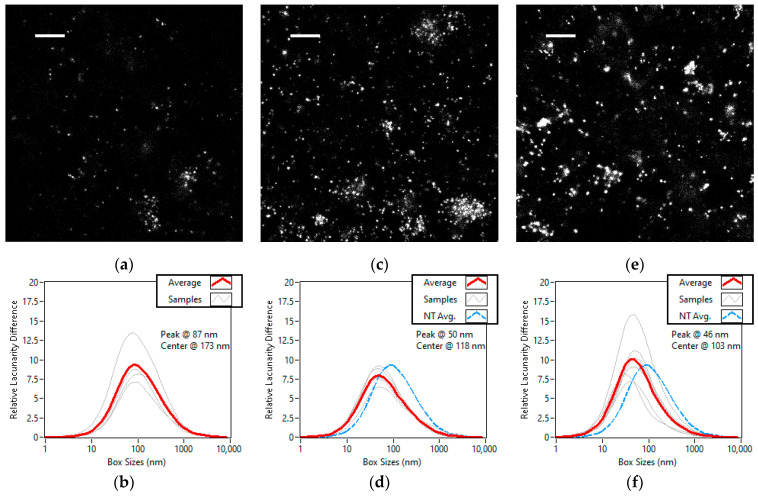
Super-resolution images and lacunarity difference curves of A647 labelled γH2AX in the nuclei of U2OS and DIvA cells grouped by treatment. Untreated U2OS (**a**,**b**), NCS treated U2OS (**c**,**d**) and 4-OHT treated DIvA (**e**,**f**) cells. Scale bars are 1 µm. The number of studied cells is four, five, and six, respectively. On each lacunarity difference curve, the average of the curves for each cell is shown in red and the average of the untreated U2OS is shown in dashed blue.

**Table 1 cells-11-03105-t001:** Table showcasing the values of the simulation parameters and their respective ranges.

Names of the Simulation Parameters	Base Values of the Simulation Parameters	Ranges of the Simulation Parameters
Cluster number	9	1–9
Cluster size (nm)	560	140–980
Cluster distance (nm)	2500	500–2500
Nanofocus density (nanofoci/µm^2^)	40	10–70
Nanofocus Size (nm)	55	11–99
Localizations per nanofocus (localizations/µm^2^)	150	10–290
Nonspecific localization density (localizations/µm^3^)	70	0–560

## Data Availability

Not applicable.

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
