# Peer review of "Application of Lacunarity for Quantification of Single Molecule Localization Microscopy Images"

_cells, 2022, doi:10.3390/cells11193105_

Round 1
Reviewer 1 Report
The manuscript by Kovacs et al. describes a novel and interesting approach in SMLM for quantitative analysis of localization data. The authors employ lacunarity, which is a method for measuring patterns in fractal geometry.
The principle of lacunarity as used in SMLM is well described, the obtained results are carefully analyzed by means of simulated data and biological data is finally analyzed using previously described models.
The authors highlight the speed at which quantitative data analysis can be performed using lacunarity and compare their results with findings from alternative cluster analysis approaches, i.e. DBSCAN.
Overall, the manuscript is well written and interesting to read. The paper is well suited for publication in Cells and I only have some minor comments that should be addressed before publication.
- line 41: better use "overcome" than "break"
- p.3f: px size Epsilon: Please mention already here what a typical pixel size would be in nanometer.
- p.5: it might be good to mention that the peak box size does not correspond to cluster size
- line 444: "In the second category for nanofocus density, the five studied settings were 10/μm2, 25/μm2, 40/μm2, 55/μm2 and 70/μm2.", mention localization density or localizations / μm2 at some point.
- p. 5f: To avoid confusion, the terms nanofocus density and nanofocus size should be carefully described. What is a nanofocus in the context of a cluster?
- Figs. 5-7: what is center referring to (e.g. center @ 136 nm)?
- l.299 "The peak of the lacunarity difference curve moves towards the larger box sizes with an increase in homogeneity", smaller box sizes?
- As mentioned in the discussion, DBSCAN provides similar results. It might be helpful for the reader if the author show this in their manuscript/Supporting Information with data processed by the two methods.
- Would the method work on the level of reconstructed images (e.g. with 5 nm pixel size) or is it preferable to work on raw localizations?
Reviewer 2 Report
In this manuscript, Kovács et al describe an image analysis method for extracting molecular clustering information from single-molecule localization microscopy (SMLM) data. The method comes from the analysis of lacunarity which has been used in other fields, and the authors tested it with simulated SMLM images with clusters and experimental data of γH2AX in DNA repair. It was found that lacunarity is sensitive to the properties of both the clusters and the “nanofoci”. The manuscript is overall clear and readable, yet some essential method details are missing. Despite that the application of lacunarity analysis to SMLM is novel and potentially interesting to the community, I have serious concerns about the data interpretation and practical significance of this method for cluster analysis.
Major points:
1. Conceptually, the lacunarity difference (LD) curve reflects cluster properties in two dimensions: the peak box size and the peak amplitude. However, as the authors showed, at least six cluster parameters could affect the LD curve (Fig. 4e). This makes the interpretation of LD curve ambiguous, as a trend can be a result of combination of multiple factors altered. For example, the changes from Fig. 5a to 5b could as well be interpreted as combination of nanofoci density increase and nanofoci size decrease instead of the author’s interpretation of changes in cluster size. Therefore, in my opinion, the LD method does not really quantify the cluster properties as DBSCAN does. The findings on experimental data do not appear to go beyond what can be seen by naked eye.
2. The simulation parameters are poorly described. Did the authors keep all other parameters constant? For example, were nanofoci density kept the same between Fig. 2d-f? The cluster sizes also appear different between Fig. 2g-i. It would be helpful to make a table listing all the parameters in simulation. In addition, a “no cluster” control is missing. How would the LD curve change when the nanofoci numbers are kept the same, but they do not form clusters? This is important because it seems that the LD curve is mostly reflecting the nanofocus size (Fig. 3k).
3. According to Fig. 1, the normalization is performed by comparing the LD curve to a random distribution. Is this a random distribution of localizations or nanofoci? In the formation of SMLM image, each protein (or proteins localized within the resolution limit) will give rise to multiple localizations, making a nanofocus. In the case of a single protein giving rise to a nanofocus, it does not make sense to disperse the localizations randomly as normalization. Because normalization is pivotal to the LD curve, this process must be described in detail.
4. The only advantage of this method might be its faster speed than other clustering methods such as DBSCAN. However, there is no quantitative comparison between the two methods in the manuscript in terms of speed or result. Throughout the manuscript, the authors tend to interpret the LD curve changes because of image homogeneity change. This is intuitive, but it also raises the question why the authors did not choose other algorithms for image homogeneity analysis, which are potentially even faster than the LD method.
5. As an image analysis paper, it is frustrating that all images are missing scale bars or scale bar annotations. The rendering of images is also inconsistent, such as Fig. 5a and Fig. 6c, which are essentially the same but rendered differently. The presentation parameters of simulation data also need to be consistent.
Minor points:
1. Line 46-47: depending on how people define SMLM, the recently developed MINFLUX method has achieved the highest resolution for fluorescence nanoscopy.
2. Line 50: for PAINT, please cite the original Sharonov & Hochstrasser paper instead.
3. Line 83: need citation here.
4. The figure legends are poorly arranged. To improve readability, please describe them alphabetically and add a summary sentence at the beginning of each figure.
5. Figure 1: (e) and (f) do not match in amplitude.
6. Line 172: should be γH2AX.
7. Line 300: “smaller” box size?
8. Line 300: this interpretation is simpler but may not be real. See discussion of major point 1.
Reviewer 3 Report
The manuscript titled 'Application of lacunarity for quantification of single molecule localization microscopy images' adopts a unique analytical approach to demonstrate the application of lacunarity to localization datasets. The authors have systematically laid the framework for the approach and verified its successful application using previously published and verified datasets. The parallel comparison approach helps ascertain the significance of the algorithm presented within the manuscript. The manuscript is well written with systematically performed experimentation. The approach is of significant importance to the field of SMLM analysis.
Author Response
Dear Reviewer,
Thank you for reviewing our manuscript.
Round 2
Reviewer 2 Report
The authors did a great job addressing my comments. I thus recommend its publication.